# The Immune Landscape of Colorectal Cancer

**DOI:** 10.3390/cancers13215545

**Published:** 2021-11-04

**Authors:** Artur Mezheyeuski, Patrick Micke, Alfonso Martín-Bernabé, Max Backman, Ina Hrynchyk, Klara Hammarström, Simon Ström, Joakim Ekström, Per-Henrik Edqvist, Magnus Sundström, Fredrik Ponten, Karin Leandersson, Bengt Glimelius, Tobias Sjöblom

**Affiliations:** 1Department of Immunology, Genetics and Pathology, Uppsala University, 75185 Uppsala, Sweden; patrick.micke@igp.uu.se (P.M.); max.backman@igp.uu.se (M.B.); Klara.hammarstrom@igp.uu.se (K.H.); simonstrom96@gmail.com (S.S.); joakim.ekstrom@igp.uu.se (J.E.); per-henrik.edqvist@igp.uu.se (P.-H.E.); magnus.sundstrom@igp.uu.se (M.S.); fredrik.ponten@igp.uu.se (F.P.); bengt.glimelius@igp.uu.se (B.G.); tobias.sjoblom@igp.uu.se (T.S.); 2Department of Oncology-Pathology, Cancer Center Karolinska, Karolinska Institutet, 17164 Stockholm, Sweden; alfonso.martin.bernabe@ki.se; 3City Clinical Pathologoanatomic Bureau, 220116 Minsk, Belarus; inahrinchyk@gmail.com; 4Department of Translational Medicine, Lund University, 20502 Malmö, Sweden; karin.leandersson@med.lu.se

**Keywords:** colorectal cancer, multiplex, tumor immunology, immune landscape

## Abstract

**Simple Summary:**

We sought to provide a detailed overview of the immune landscape of colorectal cancer in the largest study to date in terms of patient numbers and analyzed immune cell types. We applied a multiplex in situ staining method in combination with an advanced scanning and image analysis pipeline akin to flow cytometry, and analyzed 5968 individual multi-layer images of tissue defining in a total of 39,078,450 cells. We considered the location of immune cells with respect to the stroma, and tumor cell compartment and tumor regions in the central part or the invasive margin. To the best of our knowledge, this study is the first comprehensive spatial description of the immune landscape in colorectal cancer using a large population-based cohort and a multiplex immune cell identification.

**Abstract:**

While the clinical importance of CD8+ and CD3+ cells in colorectal cancer (CRC) is well established, the impact of other immune cell subsets is less well described. We sought to provide a detailed overview of the immune landscape of CRC in the largest study to date in terms of patient numbers and in situ analyzed immune cell types. Tissue microarrays from 536 patients were stained using multiplexed immunofluorescence panels, and fifteen immune cell subclasses, representing adaptive and innate immunity, were analyzed. Overall, therapy-naïve CRC patients clustered into an ‘inflamed’ and a ‘desert’ group. Most T cell subsets and M2 macrophages were enriched in the right colon (*p*-values 0.046–0.004), while pDC cells were in the rectum (*p* = 0.008). Elderly patients had higher infiltration of M2 macrophages (*p* = 0.024). CD8+ cells were linked to improved survival in colon cancer stages I-III (q = 0.014), while CD4+ cells had the strongest impact on overall survival in metastatic CRC (q = 0.031). Finally, we demonstrated repopulation of the immune infiltrate in rectal tumors post radiation, following an initial radiation-induced depletion. This study provides a detailed analysis of the in situ immune landscape of CRC paving the way for better diagnostics and providing hints to better target the immune microenvironment.

## 1. Introduction

Cancer remains one of the leading causes of death worldwide and CRC is the third most common cancer type and the second most common cancer killer [1]. In addition to the traditional TNM classification system, molecular subgroups based on mutations and gene expression profiles are used to identify more homogeneous subgroups as CRC is intrinsically heterogeneous [2]. In particular, somatic mutations in driver genes, such as those of the RAS pathway, have major clinical implications for the response to specific therapies and molecular testing for such mutations is now a clinical standard in metastatic CRC (mCRC). During the past decade, a classification system based on the tumor immune environment has attracted attention. Galon et al. introduced an immune score that grouped CRCs with regard to the infiltration of T cells (CD3+ and CD8+ lymphocytes) in the tumor and the invasive margin [3]. The Immunoscore^®^ provides independent prognostic information in addition to other clinical parameters including the TNM classification in CRC stage I–III [4,5]. Furthermore, not only the T cell lineage, but also the presence of other immune cell types including B cells and NK cells have been associated with better outcomes [6,7]. On the other hand, certain immune contexts of the primary tumor dominated by immune suppressive cells, like T regulatory cells or M2 type macrophages, were connected to tumor progression and poor prognosis [8]. These observations indicate an active involvement of the tumor immune environment in tumorigenesis and suggest a diagnostic, prognostic, and, potentially, also predictive value of a deeper immune classification of CRC.

The introduction of immune checkpoint inhibitors has demonstrated that cancer immunity can be modified, leading to immune-mediated long-lasting tumor regression in subsets of patients with several different solid tumor types [9]. Further, the pre-existing microenvironment seems of major relevance and high infiltration with immune cells is associated with better tumor response and long-term survival in patients treated with checkpoint inhibitors [10]. Transcriptomic analyses have revealed that these tumors also express inflammatory and effector cytokines, indicating a basic anti-tumor immune response, though not efficient enough to control tumor growth. This immune phenotype has been designated ‘inflamed’ or ‘hot’. In contrast, tumors with less immune cell infiltrate were designated as ‘desert’ or ‘cold’ tumors [11,12]. In CRC, the ‘inflamed’ immune phenotype is often found in tumors with high microsatellite instability (MSI-H), most probably due to high tumor mutational load and the presentation of neoantigens leading to anti-cancer immunity [13,14]. The analysis of tumor exomes allows identification of such neoantigens. The number of mutations per exome ranges from ~100 in microsatellite stable (MSS) to ~1000 in MSI CRC [15,16,17]. Checkpoint inhibitor therapy is effective in these tumors [18,19] and is now approved for mCRC with MSI-H [18,19,20,21]. Taken together, there is evidence that the tumor immune microenvironment plays a major role in terms of CRC prognosis and, at the same time, indicates whether immune modulating treatment is beneficial.

Despite its obvious clinical relevance, knowledge of the immune microenvironment in CRC is fragmentary as most studies have focused on only a single cell type or a few subsets of immune cells. The most applied strategy is based on immunohistochemical analysis with semi-quantitative measurements, carrying a substantial risk of observer bias. Multiple markers may be analyzed in consecutive sections, but this has limited relevance when evaluating cell interactions [22]. Therefore, the focus of prior immunohistochemical studies was on the T cell lineage. More comprehensive studies rely on deconvolution of gene expression data, without spatial context of immune cells. This approach has disadvantages, as low abundance cell types are challenging to quantitate accurately in bulk mRNA profiles. Given these methodological difficulties, there are few comprehensive efforts towards in situ mapping of the tumor microenvironment [23]. However, novel immunofluorescence multiplex techniques in combination with advanced scanning and image analysis systems can tackle these obstacles to describe the immune response in cancer in a holistic and standardized manner [24].

The aim is to apply immunofluorescence multiplexing techniques to provide the first comprehensive overview of the immune landscape across a large population-based cohort of CRC patients. Relevant molecular and clinical subgroups are analyzed using antibody panels allowing in situ identification of 15 distinct subclasses of immune cells in association with clinical parameters and outcome. Finally, we compare the immune status in rectal tumors treated with different therapies and intervals prior to surgery to identify therapy-induced modulation of CRC immunity.

## 2. Materials and Methods

### 2.1. Study Cohort

The study cohort consists of prospectively collected CRC patients living in the Uppsala region of Sweden, most of which have been included in the Uppsala-Umeå Comprehensive Cancer Consortium (U-CAN) [25]. In total, 937 patients were diagnosed with CRC between 2010 and 2014 in the Uppsala region. Of them, 746 (80%) were included in tissue microarray (TMA). For the present study, only patients with TMA material from primary tumors were selected. After the staining procedures and quality control, 536 patients were available for analysis. The clinicopathological characteristics of the included patients and their tumors are presented in Appendix A.

All patients received stage-stratified standard of care according to the Swedish national guidelines from 2008. According to the guidelines, colon tumors were recommended primary surgery and adjuvant chemotherapy if risk-factors for recurrence were present. If the colon tumor was considered inoperable, neoadjuvant chemotherapy was administered to shrink the tumor before surgery. Rectal tumors were grouped into three prognostic categories: early (low recurrence risk), intermediate (intermediate recurrence risk), and locally advanced (high recurrence risk) with recommendations of primary surgery or pre-operative radiotherapy or chemoradiotherapy with different time-intervals to surgery, dependent on group belonging. Formalin-fixed paraffin-embedded tissue blocks of primary tumors and distant metastases were used to construct TMAs. Each case was represented on the TMA with cores derived from the central part of the tumor and from the invasive margin. The study was approved by the regional ethical committee in Uppsala, Sweden (Dnr 2010/198 and Dnr 2015/419).

MSI status was evaluated in available cases by IHC analysis with antibodies against the two MMR proteins, PMS2 and MSH6. The tumor was denoted as MSI-H if at least one of these proteins was absent.

### 2.2. Multiplex Immunofluorescence Staining

For the multiplexed immunofluorescence staining, 4 µm thick sections were de-paraffinized, rehydrated, and rinsed in distilled H_2_O. Three staining protocols were established with three panels of antibodies: a lymphocyte panel, with CD4, CD8, CD20, FoxP3, CD45RO, and pan-cytokeratin (pan-CK) (as described in [26]); a NK/macrophage panel encompassing CD56, NKp46, CD3, CD68, CD163, and pan-CK; and a dendritic cell panel with CD3, CD1a, CD208, CD123, CD68, CD15, and pan-CK. The staining procedure was performed as described [27,28]. In total, 520 cases were evaluable for the lymphocyte panel, 508 cases for the NK/macrophage cell panel, and 498 cases for the dendritic cell panel (Appendix A). Using a combination of immune markers, we quantified 15 immune cell subclasses (Figure 1a,b).

### 2.3. Imaging, Image Analysis, Thresholding and Immune Scores

The stained TMAs were imaged using the Vectra Polaris system (Akoya Biosciences, Marlborough, MA, USA) in a multispectral mode at a resolution of 2 pixels per μm. This resulted in 5968 individual multi-layer images, each representing a TMA core. Spectral deconvolution and initial image analysis were conducted in the inForm (2.4.6) software (Akoya Biosciences) (Appendix A). Each of the images was reviewed and manually curated by a pathologist to exclude artefacts, staining defects, and accumulation of immune cells in necrotic areas and intraglandular structures. The vendor-provided machine learning algorithm was trained and applied to split tissue into three categories: tumor compartment, stromal compartment, or blank areas as described [29]. Cell segmentation was performed using DAPI nuclear staining as described [27,28]. The perinuclear region at 3 μm (6 pixels) from the nuclear border was considered the cytoplasm area. The nuclear or cytoplasmic area was evaluated for the expression of nuclear or cytoplasmic/membrane markers, respectively. The cell phenotyping function of the inForm software was used to manually define cells positive to each of the markers. The intensity of the marker expression in selected cells was used to set the thresholds for marker positivity. The defined thresholds were then applied to the raw output data of the complete cohort outside the inForm pipeline. Every cell was characterized as positive or negative for each marker in the panel, and marker co-expression was used to define immune cell subtypes (Figure 1a,b). Immune cell infiltration was evaluated as the number of cells per analyzed tissue area, in the stromal compartment and tumor compartment. This algorithm was applied to quantify the 15 different immune cell subclasses in the stroma and tumor compartment in the center of the tumor and in the invasive margin, i.e., obtaining a cell quantification in four tissue regions. Immune scores were generated for each immune cell subclass. First, immune cell infiltration in each of the four localizations was dichotomized into 0 (low) and 1 (high), using the median as threshold. The sum of the values gives a score between 0 and 4 (Appendix A).

### 2.4. Statistics

Statistical analyses were performed using R (version 3.5.1) and SPSS V20 (SPSS Inc., Chicago, IL, USA). In radically operated stage I–III patients, recurrence-free survival (RFS) was computed as the time from surgery to the first documented disease progression including local recurrence or distant metastases or death due to any reason, whichever occurred first [30]. Overall survival (OS) was the time from surgery to death due to any reason. To estimate relative hazards in both univariate and multivariable models, a Cox proportional hazards model was used. Hierarchical clustering analyses were conducted with the heatmap.plus package (version 1.3) in R. For the analyses of associations between MSS/MSI status and metastases type, the Chi-square test was used. The Ward algorithm was used for hierarchical clustering and *p* < 0.05 was considered statistically significant. The Benjamini–Hochberg procedure was used to adjust for multiple hypothesis testing, and adjusted q-values were reported.

## 3. Results

### 3.1. Identification and Quantification of Immune Cell Subclasses in CRC by Multiplex Staining

The successfully stained tissue microarray cores comprised of 536 surgically removed primary CRC cases with two cores from each tumor, representing the invasion margin and the central tumor area. Two thirds of the patients had colon cancer while one third rectal cancer, 54% of the patients were male and 35% were older than 70 years. In total, 59% of patients had stage III disease and 15% had metastatic disease at diagnosis. Most colon cancer patients (99%) were therapy-naïve at the time of surgery, while many rectal cancer patients had received pre-operative treatment (61%) (Appendix A). The TMAs were stained with three different panels of immune markers along with pan-cytokeratin and DAPI as nuclear stain. Examples of multiplex immunofluorescent images are shown in Figure 1a and the analysis pipeline is illustrated in Appendix A. The expression of different immune markers was combined to assign each cell to one of 15 immune cell subtypes (Figure 1b), including different lymphocytes, macrophages, natural killer (NK) cells, dendritic cells (DCs), and myeloid cells (Figure 1b). The density of immune cell subtypes was annotated in the stroma and tumor compartments of the tumor center and at the invasive margin, resulting in four different metrics for each immune cell class. Overall, the most abundant cell types were CD8 single positive cells and M1 macrophages with median (mean) values of 314 (832) and 431 (685) cells per mm^2^, respectively. NK cells and NKT cells demonstrated very low overall density with 77 and 81% of the cases being negative, respectively (Figure 1c,d). Taken together, the infiltration of immune cells was highly variable between tumors and immune cell subclasses, spanning from 0 to 11,994 cells/mm^2^.

### 3.2. Spatial Distribution of Immune Cells in CRC

Next, we performed case-wise comparisons between the four tissue regions (stroma and tumor in the tumor center and stroma and tumor in the invasion margin). When comparing infiltration in stroma against tumor compartments (Figure 1d and Appendix A), most immune cell subsets were more abundant in the stroma. Only CD8 single positive cells (see Figure 1B for immune cell sub-classification) and myeloid cells were more numerous in the tumor compartment (q < 0.001, Mann–Whitney U test with Benjamini–Hochberg correction). The distribution of immune cells between the center of the tumor and the invasive margin were similar, with a few notable exceptions. The most striking difference was observed for T cells, which were more abundant in the tumor center (Appendix A). In conclusion, there was greater enrichment of immune infiltrate in the stroma compared to the tumor cell compartment, but no significant differences between tumor center and invasive margin.

### 3.3. Interrelationship of Immune Cells and Immune Scores

We hypothesized that immune cells of the same lineage infiltrate tumor tissue in a coordinated fashion. Therefore, we correlated the abundance of all immune cell subtypes to each other in the four analyzed tumor regions (Appendix A). Indeed, the correlations for each specific immune subclass between the four tissue regions of the same tumor sample were in generally high. Due to this observation, we summarized the immune cell values in a single score for each immune cell subclass. These scores were generated in analogy to the original Immunoscore^®^ [3] by summarizing the cell densities in all four regions into one score ranging from 0 to 4 (Appendix A). Analysis using the immune scores revealed interrelations between different immune cells, with the highest correlations between lymphocyte subtypes, and between M1 macrophages and CD8+ lymphocytes. Interestingly, NK and NKT cells correlated negatively to mature dendritic cells and plasmacytoid dendritic cells (mDCs and pDCs). There was also a negative correlation of T cells to myeloid cells and M2 macrophages (Figure 2a). In conclusion, we identified distinct dominating immune infiltration patterns when a set of immune cell subclasses infiltrate tumor tissue coordinately.

### 3.4. Clustering of CRC Cases by Immune Cell Scores and Relation to Clinical Parameters

We next evaluated whether the immune scores were related to clinicopathological parameters. The findings largely replicated associations observed in region-restricted immune cell densities (Appendix A). In line with published data [31,32], tumors of the right colon were characterized by higher immune scores for most T cell subclasses, M2 macrophages, and myeloid cells in comparison to the left colon and rectum, while pDC cells were enriched in the rectum. The most abundant immune infiltrates were seen in the tumors from flexura hepatica and colon transversum (Figure 2b and Appendix A). Higher immune scores of CD8 single positive cells were observed in tumors with lower N stage. Most T cell subclasses, macrophages, and myeloid cells were enriched in MSI-H tumors. The M2 macrophages were associated with higher patient age.

To capture the dominating immune landscapes, we performed hierarchical clustering based on immune scores across all 373 therapy-naïve cases. The cluster analysis revealed two distinct groups (Figure 2c). The smaller cluster (*n* = 145) included tumors with high immune scores for T cells, reflecting an ‘inflamed’ phenotype. Interestingly, this cluster consisted of two distinct subgroups with high CD8 or CD4 scores. The second, larger cluster (*n* = 228), demonstrated low immune scores for T cells, representing the immune ‘desert’ phenotype. Within this cluster, several smaller subgroups were observed with either increased M2/myeloid cell scores, NK/NKT cell scores, immature dendritic cell (iDC) scores, or mDC/pDC scores. The ‘inflamed’ cluster was enriched with tumors (i) from the right colon, (ii) with high differentiation grade, (iii) without neural invasion, and (iv) with MSI. Other parameters, such as stage, sex, vascular engagement, local lymph node involvement, presence of distant metastases, or BRAF mutation status did not affect the distribution across the main clusters (Appendix A). Interestingly, when we analyzed the impact on OS, the ‘inflamed’ and ‘desert’ immune clusters did not demonstrate significant differences. Taken together, tumors with an ‘inflamed’ or a ‘desert’ immune phenotype were clearly distinguishable, although, unexpectedly, not associated with improved or reduced survival.

### 3.5. Immune Scores and Survival

Clearly defined ‘hot’ and ‘desert’ tumors did not have significant survival differences. Therefore, we hypothesized that individual variations in different immune cell subsets may play more important roles in predicting patient survival and focused on the analyses of single immune scores. In a first set of survival analyses, we evaluated OS in all therapy naive patients (Figure 3a and Appendix A); since preoperative treatment may influence the immune scores, these analyses were restricted to untreated patients. In line with previously published data [32], T cell immune scores had positive associations with improved survival, but only the immune score for CD8 single positive cells reached statistical significance (HR = 0.64, 95%CI [0.49–0.84], q = 0.014). In contrast, higher M2 macrophage scores were associated with shorter survival (HR = 1.50, 95%CI [1.20–2.00], q = 0.014). Due to the heterogeneity of CRC, both in terms of natural course of the disease and treatments, we investigated survival in specific patient subgroups with relevant endpoints. The same survival impact of CD8 cells and M2 macrophages was seen for radically operated stage I–III colon cancer patients, when disease-free survival (RFS) was analyzed. However, in the multivariable analysis, adjusted to clinicopathological factors, only single positive CD8 cells had a significant impact on prolonged RFS (HR = 0.64, 95%CI [0.41–0.98], *p* = 0.039) (Figure 3b and Appendix A). Subsequently, we evaluated stage IV patients separately. Immune scores for single positive CD4, single positive CD8 cells, and mDCs were associated with longer survival in the univariable analyses, although only the first retained statistical significance after adjustment for multiple testing (Figure 3c and Appendix A). Thus, survival analysis in a therapy-naïve cohort and in colon cancer stage I–III confirmed previous findings indicating a major impact of CD8+ cells. In stage IV, single positive CD8+ were accompanied by mDCs and even stronger survival-predictive impact of single positive CD4+ cells.

### 3.6. Rectal Cancer

Rectal and colon cancer are often considered separate diseases [33]. This is also reflected by the different immune phenotype observed with lower numbers of CD4 single positive cells, CD4 Treg cells, and higher mDC, pDC, NKT cells in therapy-naïve rectal cancer compared to colon cancer (Appendix A, Figure 4a), suggesting a lower level of natural immune activation in rectal cancer patients. Since most rectal cancer patients receive neoadjuvant radiotherapy or chemoradiotherapy (RT/CRT), we analyzed samples from 78 patients treated with RT/CRT. These patients were dichotomized regarding neoadjuvant treatment type, that was either (i) short-course RT (5 × 5 Gy in one week) followed by immediate surgery or (ii) short-course RT with delayed surgery (later than three weeks), CRT with delayed surgery, or short-course RT and chemotherapy in the interval to surgery. The analyses revealed that many immune cell counts decreased in the group that received RT/CRT therapy and was operated soon after the treatment and increased again in the delayed surgery group (Figure 4b), with the most characteristic profile seen for CD4+CD45RO+, CD8+CD45RO+, CD8 regulatory cells, M2 macrophages, iDCs, and pDCs. Interesting, CD4 and CD8 single cells, as well as B cells, showed quite a stable level of infiltration independent from neoadjuvant treatment type. None of the immune cell subclasses showed statistically significant differences when comparing tumors from primary surgery and those from the delayed surgery group, with the exception of M1 macrophages which demonstrated lower densities in the pretreated delayed surgery group. Taken together, the immune profiles differ between rectal and colon cancers in several aspects. In rectal cancer, the pattern reflects repopulation of the immune infiltrate in tumor tissue post radiation, following an initial radiation-induced depletion.

## 4. Discussion

Tumors are composed of malignant cells and host elements of the tumor microenvironment which can support or suppress tumor progression and influence anti-cancer treatment. Although T cells have been considered as the most important anti-tumoral immune cells, detailed analysis of T cell subtypes and of other immune cells classes has been limited due to methodological difficulties. The functions of different immune cells can vary dramatically, depending on their activation and differentiation status. These differences are reflected in unique protein expression profiles, requiring techniques for multiplex in situ analysis to enable quantification of immune cell classes in clinical samples [34]. This study describes the immune cell microenvironment of CRC with 15 subgroups of immune cells at a hitherto unrivaled resolution. We applied a multiplex in situ staining method in combination with an advanced scanning and image analysis pipeline, akin to flow cytometry in situ, and analyzed 5968 individual multi-layer images of tissue defining a total of 39,078,450 cells. Each image was reviewed and thoroughly curated by a pathologist to exclude artefacts, staining defects, and necrotic areas. Furthermore, we considered the location of immune cells with respect to the stroma and tumor cell compartment as well as tumor regions in the central part or the invasion margin. To the best of our knowledge, this study is the first comprehensive spatial description of the immune landscape in CRC using a large population-based cohort and a multiplex immune cell identification.

In addition to commonly analyzed immune cells, like CD4 and CD8 cells, or FoxP3+ cells, we could accurately discriminate additional subsets of T lymphocytes. This increased the depth of cell sub-classification and, at the same time, improved the purity of each cell class. For instance, in conventional immunohistochemical analysis, FoxP3 positive cells have usually been considered regulatory T cells. Our approach refined cell counting by excluding FoxP3+ cells of non-lymphocyte or unknown origin, e.g., cancer cells or immune cells negative for other markers [28,35]. Furthermore, we found that a large proportion of the FoxP3+ cells are of the CD8+ lineage. CD8+FoxP3+ T cells have previously been suggested to be immunosuppressive CD8+ Tregs [36], although conflicting data exist showing that FoxP3 may be induced upon CD8+ T cell activation [37]. The unexpected high abundance of the specific cell type observed here should be the subject of further investigations.

We could clearly identify two distinct immune phenotypes: immune ‘inflamed’, characterized by high infiltration of lymphocytes, and immune ‘desert’ tumors. Interestingly, the ‘inflamed’ cluster in our analysis consisted of two subgroups with tumors with either CD4+ or CD8+ infiltration, and with only a small group of cases with concurrent high CD4 and CD8 levels. One may speculate that this finding might explain the general resistance of CRC to immune checkpoint inhibitors, considering that the presence of both immune cell linages is necessary for effective cancer cell elimination. Finally, the presence of dendritic cells, cells of the myeloid lineage, and NK cells define further subgroups within the immune desert background. Taken together our analyses refine the immune classification of CRC.

Despite being expectedly associated with dMMR/MSI-H cases, immune ‘inflamed’ tumors did not demonstrate a statistically significant association with improved patient survival. We next extended our analysis of individual immune cells in the context of clinical outcome. In an objective and unbiased analysis, previously reported relations were confirmed, but we could also uncover new information about the prognostic impact of further immune cell subclasses. Thus, Immunoscore^®^, which considers amount and localization of CD3 and CD8 cells showed an independent prognostic impact in a large multicenter prospective study [5]. Our CD8 immune score, although generated slightly differently from the Immunoscore^®^, had prognostic value in this cohort. Another immune cell subclass which emerged as a potent prognostic biomarker was the M2 macrophages. These cells have a broad and not yet fully understood role, but can be considered as pro-tumoral elements and hallmarks of an immunosuppressive microenvironment [38]. Our findings suggest that the adverse effect of M2 macrophages should be considered in efforts to improve the prognostic accuracy of immune scoring systems.

Finally, we evaluated changes in the immune microenvironment of primary rectal cancer tissue subjected to preoperative RT/CRT. Our results demonstrated immune deprivation in tumor tissue undergoing resection directly after irradiation. The local immunosuppressive effect of irradiation is well established, and diverse radio-sensitivity of different immune cell types has been reported (reviewed in [39]). In agreement with these reports, we observed lower cell counts for all evaluated cell types. While there are several studies describing the immediate effect of RT/CRT on the tumor microenvironment, data about delayed effects, after weeks or months, are largely missing. Here, tumors resected after a delay following RT or CRT were characterized by an immune microenvironment largely similar to non-irradiated tumors. Accordingly, the immune suppressive effect of therapy should be considered when combinations of immune and conventional therapy are planned and may give a rationale for the sequencing of different therapy modalities in clinical trials. However, this simplistic explanation is complicated by the fact that the tumors that received or did not receive neoadjuvant treatment were not randomized, but rather selected according to stage and other characteristics on magnetic resonance imaging and type and intensity of therapy varied. The patients of the three analyzed groups, i.e., primary surgery, preoperative RT followed by immediate surgery, and preoperative RT/CRT followed by delayed surgery, are not comparable with regards to clinicopathological characteristics. With this caveat, the causes for the reported observations may be more elaborate than a direct link between RT/CRT and immune cell count. Overall, early (or so called ‘good’) rectal tumors [40] were operated immediately and had lower stages and few other risk factors (like extramural vascular invasion) than intermediate or ‘bad’ tumors subjected to RT and immediate surgery. Further, the group of locally advanced tumors receiving preoperative RT/CRT and delayed surgery (so called ‘ugly’ tumors) usually represent even more advanced tumor forms: stage cT4a/b or cT3 tumors with threatened/involved mesorectal fascia. Taking this together, the immune characteristics can not only be compared between neoadjuvant treatment groups, but also need to be normalized to their non-pretreated counterparts, with respective T and N-stages. With this background, using data presented in Appendix A as reference, one could expect ‘ugly’ tumors to have lower levels of immune infiltration. However, our data demonstrate that ‘ugly’ tumors after RT/CRT were immunologically comparable with non-pretreated ‘good’ tumors for most of the immune cells (except M1 macrophages). Therefore, we hypothesize that RT/CRT may convert ‘ugly’ tumors into immunologically ‘good’ ones. Although intriguing, this interpretation should be considered with caution because the number of cases is relatively small and due to the absence of proper non-treated reference tissue for RT/CRT cases. Further studies, involving patients randomized with regards to pre-operative treatment (in terms of the type of treatment and of the timing prior surgery) are therefore warrantied. Ideally, such studies should include sampling before the neoadjuvant treatment.

## 5. Conclusions

In conclusion, to the best of our knowledge this study is the largest in terms of patient numbers and analyzed immune cell subclasses in CRC. We provide a detailed un-biased overview of the in situ immune landscape of CRC and were able to confirm but also extend the concept of cancer immunity. Many of the observations may have clinical relevance for CRC patients by paving the way for better cancer diagnostics or by providing hints to better target the immune microenvironment therapeutically. The applied multiplex technique and the analysis pipeline are applicable on common diagnostic tissue samples; therefore, it is possible that a comprehensive analysis of the immune microenvironment will become a part of the future clinical routine in the era of immunotherapy.

## Figures and Tables

**Figure 1 cancers-13-05545-f001:**
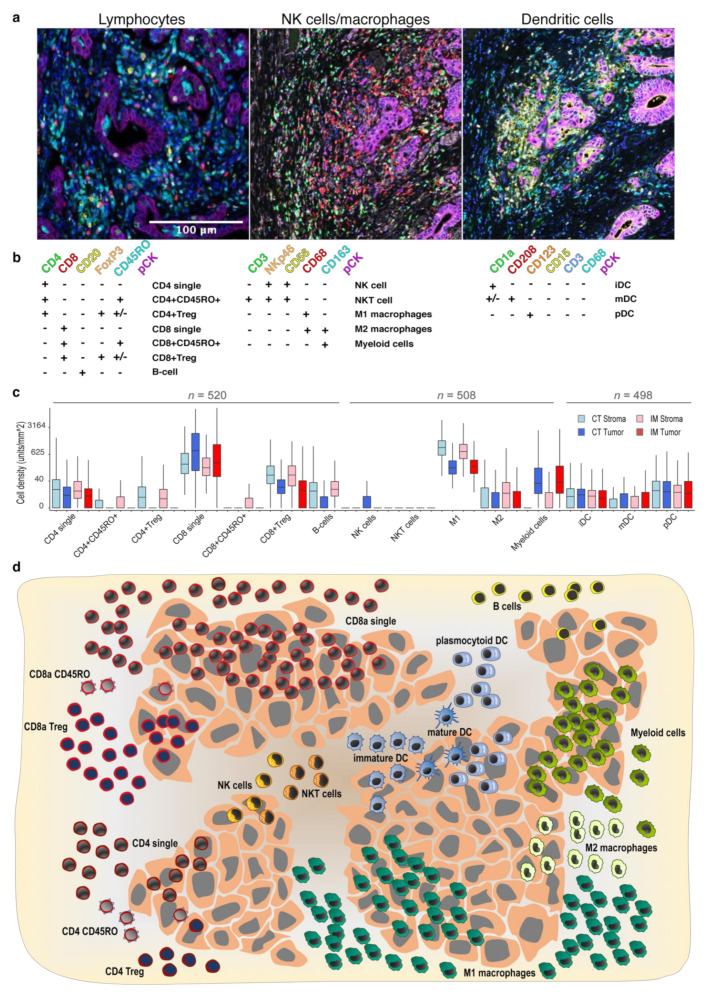
Characterization of immune cell subsets in the tumor and stroma compartments at the invasive margin and core of the tumor of primary CRC. (**a**) Representative images of the multiplex staining with three immune panels; (**b**) scheme of the immune marker combinations used to define the subgroups of immune cells; (**c**) immune cell densities in Tumor and Stroma compartments in Central Tumor (CT) and Invasive Margin (IM) (boxes show median values and interquartile range, and numbers represent cell counts per mm^2^, cube root transformed); and (**d**) illustration of the mean immune cell infiltration in tumor center in tumor and stromal compartment.

**Figure 2 cancers-13-05545-f002:**
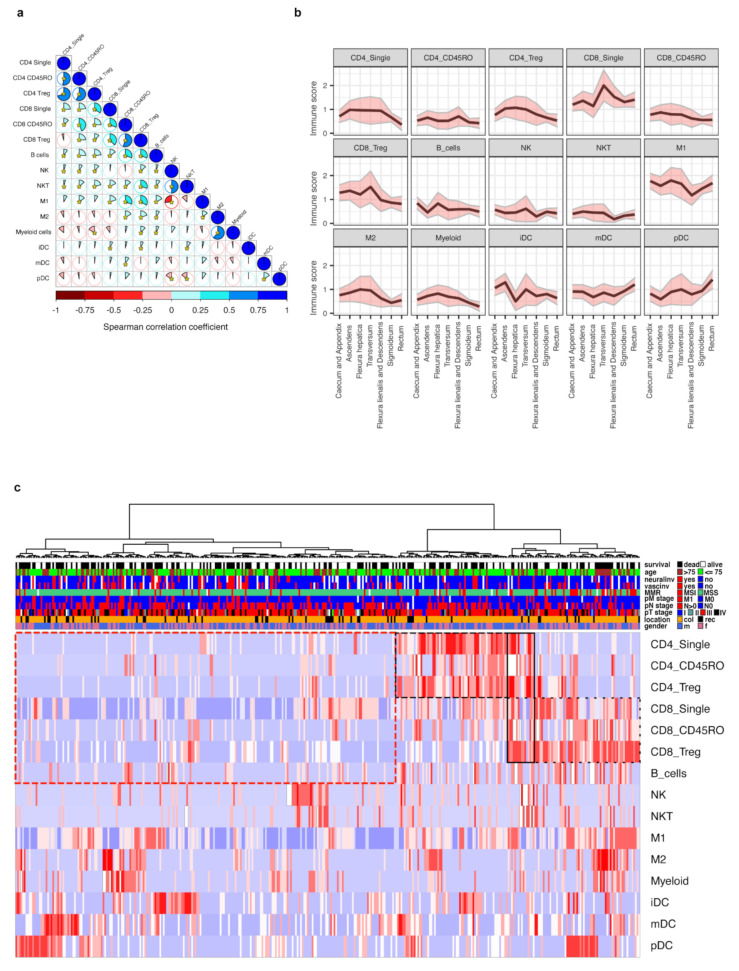
The immune scores interrelations, distribution across different clinical and pathological groups and unsupervised hierarchical clustering. (**a**) Graphical representation of Spearman’s correlation matrix between immune scores. Pie charts and the intensity of shading represent the strength of correlation (Spearman correlation coefficient), blue color indicates direct while red color indicates inverse correlation. Asterisks indicate statistical significance (*p* < 0.05). (**b**) Immune scores mean levels (black line) and 95% confidence intervals (pink areas limited by gray lines) at specific primary tumor locations. For additional data, see Appendix A. (**c**) Unsupervised hierarchical cluster analysis of immune scores. Cases were clustered based on the levels of immune scores. A total of 373 cases with complete immune score data from therapy-naïve patients were available. Clusters with enriched CD4 or CD8 cells are marked by dashed black line, while the cluster with low lymphocyte level is marked by dashed red line. For additional data, see Appendix A.

**Figure 3 cancers-13-05545-f003:**
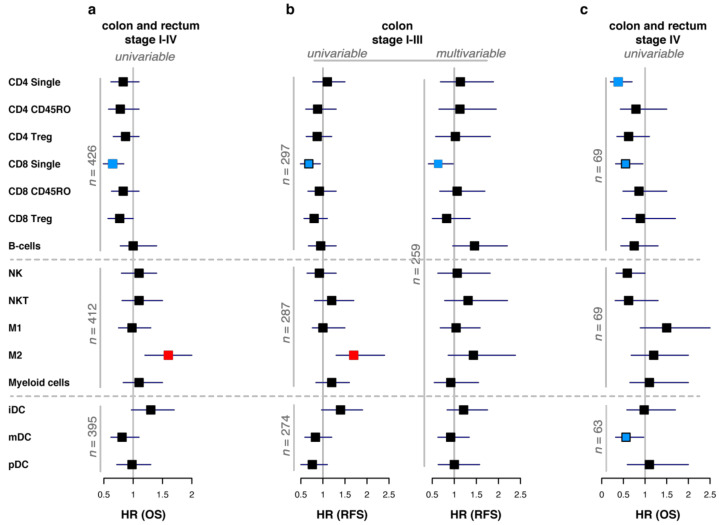
Immune scores predict patient survival. Forest plot of hazard ratios (HR) for immune scores in the univariable and multivariable Cox regression models. Filled squares indicate HR and whiskers represent 95% CI. Blue-colored squares indicate statistically significant (*p* < 0.05 and, where applicable, FDR q < 0.05) associations of the respective immune score with improved survival, while red squares represent association with reduced survival. Blue-colored squares with black contour indicate that the association was statistically significant in an individual test (*p* < 0.05) but lost statistical significance after adjustment for multiple testing (FDR q ≥ 0.05). (**a**) Univariable associations of immune scores with OS in a complete cohort of therapy-naïve patients. For detailed information see Appendix A. (**b**) Association of immune scores with RFS in stage I–III colon cancer. Left panel illustrates the result of the univariable Cox regression models. Right panel illustrates the result of the multivariable Cox regression model, adjusted to clinicopathological parameters: pT, pN stages, tumor differentiation, patient age, surgery type (elective or acute), and adjuvant treatment. For detailed information see Appendix A. (**c**) Univariable associations of immune scores with OS in stage IV therapy-naïve colorectal cancer patients. For detailed information, see Appendix A.

**Figure 4 cancers-13-05545-f004:**
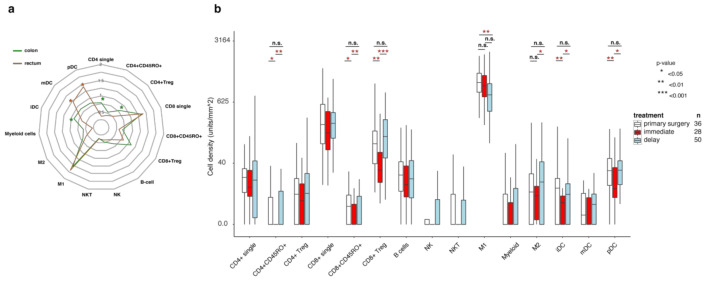
Immune infiltration in rectal cancers is restored after RT/CRT pre-treatment and delayed surgery, while vasculature is changed. (**a**) Radar plots of immune scores in therapy-naïve colon cancer patients (green) and rectal cancer patients (brown). (**b**) Immune infiltrate levels for patients who had primary surgery (white), radiation therapy followed by immediate surgery (<21 days), or delayed surgery after (chemo)radiotherapy. Numbers represent cell counts per mm^2^, cube root transformed. Boxes show median values and interquartile range of the ratios, whiskers represent 1.5 IQR. Wilcoxon signed-rank test with Pratt method assuming asymptotic distribution was used for statistical analysis. Statistically significant differences: * *p* < 0.05, ** *p* < 0.01 and *** *p* < 0.001; not statistically significant differences: n.s.

## Data Availability

Data regarding methodology, image analysis, curation and data processing, and raw data of stroma fraction are available from the corresponding author.

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
