# Peer review of "The Immune Landscape of Colorectal Cancer"

_cancers, 2021, doi:10.3390/cancers13215545_

Round 1

Reviewer 1 Report

The authors have done a remarkable work of consolidating the data regarding colon/rectal tumor patients.

The work is systematically organized and the presented.

The experimental assays done are apt and the results are presented very well.

I would like to extend my support to publish the article in your esteemed journal.

Author Response

Reviewer #1: The authors have done a remarkable work of consolidating the data regarding colon/rectal tumor patients. The work is systematically organized and the presented. The experimental assays done are apt and the results are presented very well. I would like to extend my support to publish the article in your esteemed journal.

Response: We appreciate the positive review of our study

Reviewer 2 Report

Mezheyeuski et al in their manuscript “The immune landscape of colorectal cancer” describe the immune cell microenvironment of CRC taking into consideration the MSI status. The authors performed multiplex immunofluorescence staning to identify and characterize the immune cell subclasses considering the location of the immune cells. Furthermore, the authors also provided the information about the immune classification of CRC depending on the two immune phenotypes, e.g. “hot” or “cold” cluster, and the infiltration or presence of the specific immune cell subclasses. Two immune cell subclasses, CD8 cells and M2 macrophages emerged as prognostic biomarkers. Although, immune “hot” tumors did not show statistically significant association with improved patient survival, CD8+ cells were linked to improved survival of CRC patients. The authors also reported changes in immune microenvironment of rectal cancer tissue following RT/CRT.

The described study is very comprehensive in the context of the immune cell landscape in CRC and the findings may provide insight how to better target the immune microenvironment therapeutically. The study is thorough and the manuscript is well-written. 

I only have minor comment:

Page 3: “MSI status was evaluated in available cases were evaluated by IHC analysis with antibodies against two MMR proteins PMS2 and MSH6.” The sentence is a bit unclear, so please rephrase, e.g. “In available cases, MSI status was evaluated by IHC analysis with antibodies against two MMR proteins PMS2 and MSH6.”

Author Response

Reviewer #2: I only have minor comment:

Page 3: “MSI status was evaluated in available cases were evaluated by IHC analysis with antibodies against two MMR proteins PMS2 and MSH6.” The sentence is a bit unclear, so please rephrase, e.g. “In available cases, MSI status was evaluated by IHC analysis with antibodies against two MMR proteins PMS2 and MSH6.

Response: We appreciate the comment and corrected the sentence:

“MSI status was evaluated in available cases by IHC analysis with antibodies against the two MMR proteins PMS2 and MSH6”

Reviewer 3 Report

It is my pleasure reading the article titled "The immune landscape of colorectal cancer" by Dr. Mezheyeuski et al. The paper is very well written. The study is very well designed with a clear goal to evaluate the immune microenvironment with a comprehensive spatial description of the immune landscape in a large number of colorectal cancer patients. 

There is only one main concern I would like the authors to address:

In "materials and methods" MSI stusus was evaluated in available cases were evaluated by IHC analysis with antibodies against two MMR proteins PMS2 and MSH6. MSI-H status was present if both these proteins were absent". The sentence should possibly be modified to "MSI stusus was evaluated in available cases by IHC analysis with antibodies against two MMR proteins,  PMS2 and MSH6. MSI-H status was present if both these proteins were absent". However, this methods and definition of MSI are not consistent with our understanding of the MSI. Loss of MLH1 and PMS2 are commonly seen especially if MLH1 is deficient, but MSH2 and MSH6 although makes a subunit and one protein loss may affect the other one's stability, we can still seen preserved protein expression (either MSH2 or MSH6). Meanwhile, MSI-H status does not require both MSH6 and PMS2 loss, actually this definition will significantly decrease the sensitivity identifying MSI patients. Since this study is evaluating immune landscape of colorectal cancer and MSI tumor is one important subtype in this situation, would strongly recommend the authors to review this part carefully and make modification accordingly. 

There are several minor suggestions:

  1. Related to MSI tumor, in table S1, suggest to list MSI/H with BRAF B600E mutation and MSI/H without BRAF V600E mutation since this will help us to understand the subtype better than list these two mutations separately (would imagine quite a few BRAF V600E mutated cases are MSI-H). Also suggested to update table S3 (either add rows or further subgroup the current rows).
  2. There was no explanation for the TMA abbreviation (Guess it is tissue microarray), suggest the authors to add it to help the readers understand the term. 
  3. Some minor English/Grammer check, for example, in the introduction part "The analysis of tumor exomes allows identification of such neoantigens, and the number of mutations ranges from ∼100 in microsatellite stable (MSS) to almost 1000 in MSI CRC exomes ([15] also reviewed in [16, 17])". This sentence seemed not complete. Another example was listed in the main quesition. The authors may want to review the paper and modify further.   

Author Response

Reviewer #3: It is my pleasure reading the article titled "The immune landscape of colorectal cancer" by Dr. Mezheyeuski et al. The paper is very well written. The study is very well designed with a clear goal to evaluate the immune microenvironment with a comprehensive spatial description of the immune landscape in a large number of colorectal cancer patients.

Response: We appreciate the positive review of our study

Reviewer #3: There is only one main concern I would like the authors to address:

In "materials and methods" MSI stusus was evaluated in available cases were evaluated by IHC analysis with antibodies against two MMR proteins PMS2 and MSH6. MSI-H status was present if both these proteins were absent". The sentence should possibly be modified to "MSI stusus was evaluated in available cases by IHC analysis with antibodies against two MMR proteins, PMS2 and MSH6. MSI-H status was present if both these proteins were absent".

However, this methods and definition of MSI are not consistent with our understanding of the MSI. Loss of MLH1 and PMS2 are commonly seen especially if MLH1 is deficient, but MSH2 and MSH6 although makes a subunit and one protein loss may affect the other one's stability, we can still seen preserved protein expression (either MSH2 or MSH6). Meanwhile, MSI-H status does not require both MSH6 and PMS2 loss, actually this definition will significantly decrease the sensitivity identifying MSI patients. Since this study is evaluating immune landscape of colorectal cancer and MSI tumor is one important subtype in this situation, would strongly recommend the authors to review this part carefully and make modification accordingly.

Response: We appreciate the comment. Indeed, there is a mistake in the methods description: MSI-H status was considered when one of these proteins was absent. We have reviewed the original data concerning PMS2 and MSH6 staining and can confirm that in almost all of cases only one protein was lost. In fact, there was only one exception: one tumour demonstrated PMS2 complete loss, while MSH6 showed heterogeneous expression, being positive in one cancer sample and negative in another.

As result, the description in Materials and Methods was modified:

MSI status was evaluated in available cases by IHC analysis with antibodies against the two MMR proteins PMS2 and MSH6. The tumor was denoted as MSI-H if at least one of these proteins was absent.

Reviewer #3: Related to MSI tumor, in table S1, suggest to list MSI/H with BRAF B600E mutation and MSI/H without BRAF V600E mutation since this will help us to understand the subtype better than list these two mutations separately (would imagine quite a few BRAF V600E mutated cases are MSI-H). Also suggested to update table S3 (either add rows or further subgroup the current rows).

Response: We appreciate the comment. The suggested sub-division was performed and tables S1 and S3, as well as table S2 were updated, accordingly.

Reviewer #3: There was no explanation for the TMA abbreviation (Guess it is tissue microarray), suggest the authors to add it to help the readers understand the term.

Response: The explanation of the abbreviation was added in Materials and Methods, Study cohort section.

Reviewer #3: Some minor English/Grammer check, for example, in the introduction part "The analysis of tumor exomes allows identification of such neoantigens, and the number of mutations ranges from 100 in microsatellite stable (MSS) to almost 1000 in MSI CRC exomes ([15] also reviewed in [16, 17])". This sentence seemed not complete. Another example was listed in the main quesition. The authors may want to review the paper and modify further.  

Response: The sentence was re-written:

In CRC, the ‘inflamed’ immune phenotype is often found in tumors with high microsatellite instability (MSI-H), most probably due to high tumor mutational load and the presentation of neoantigens leading to anti-cancer immunity [13,14]. The analysis of tumor exomes allows identification of such neoantigens. The number of mutations per exome ranges from 100 in microsatellite stable (MSS) to ~1000 in MSI CRC [15, 16, 17].